# Adaptation and Validation of the Spanish Version of Kogan’s Attitude toward Older People Scale (KAOP)

**DOI:** 10.3390/healthcare11091321

**Published:** 2023-05-04

**Authors:** Juan José Fernández-Muñoz, Maria Laura Parra-Fernández, Maria Dolores Onieva-Zafra, Raúl Expósito-González, José Manuel Marquinez-Rengifo, Elia Fernández-Martínez

**Affiliations:** 1Department of Psychology, Rey Juan Carlos University, 28932 Alcorcon, Spain; 2Faculty of Nursing Ciudad Real, University of Castilla-La-Mancha, 13001 Ciudad Real, Spain; 3International Doctoral School Health Sciences, University Rey Juan Carlos, 28932 Alcorcon, Spain; 4Department of Nursing, Faculty of Nursing Huelva, University of Huelva, 21001 Huelva, Spain

**Keywords:** attitudes, older people, instrument validation, gerontology

## Abstract

It is essential to understand the behavior and attitudes of nurses towards older people to improve clinical practice and quality of care in the gerontological sector. A clearer understanding of the attitudes that drive nurses toward the desire to work with older people would be a good starting point to encourage the development of positive and nurturing attitudes. A cross-sectional study with non-probabilistic sampling and a self-administered questionnaire was conducted among 381 nursing students of the Faculty of Nursing at the University of Castilla La-Mancha to evaluate the psychometric properties of the Spanish Version of the Kogan’s Attitudes Towards Older People Scale (KAOP-S). Construct validity, internal consistency, and reliability were assessed. In total, 298 females and 83 males completed the questionnaires. Their mean age was 20.42 years. The results revealed a Cronbach’s alpha coefficient of 0.75 for the scale, which is comparable with other published versions of the KAOP Scale. The results of the exploratory factor analysis established that the scale has a two-factor solution and an explained variance of 25% in the sample. The KAOP-S was found to be a reliable and valid tool with good content and construct validity for assessing nursing students’ attitudes towards older people.

## 1. Introduction

The aging population phenomenon is one of the greatest economic, social, and health challenges of the 21st century [1,2]. As new technologies advance, we find that average life expectancy also increases in those more developed countries where the quality of life in this age group has improved greatly. The population growth pyramids have been changing over the last century. At the beginning of the 20th century, triangular-shaped pyramids were still common, however, today, most European countries have a stationary pyramid profile due to the decrease in fertility and death rates together with the extensive development of science and technology. These changes have promoted the aging of the population. Because of this, most of the patients of our future health professionals will be older adults; it is an urgent need to prepare our nurses to practice nursing with the best quality care, including not only knowledge, but also skills and attitudes.

Today, the increase in life expectancy has made it necessary to create a healthy environment for older people, marked by a strategic plan from a preventive point of view, with the aim of reducing possible risks and preventing the complications and sequels of chronic diseases that surround aging. This implies an increase in the demand for primary care services, which are more profitable and accessible to the older population, and consequently, there is an increase in the responsibility of health care providers who are at the first level of care [3]. The high prevalence of illness and disability among older people leads to a high demand for care in this population group, which is mostly provided by nurses, and makes older people the main consumers of the health system [3,4].

The quality of care for older patients relies on the healthcare personnel who provide the care [5]; therefore, their attitudes influence the treatment applied to the patients, turning negative attitudes into barriers to the correct diagnosis and monitoring of older people [6]. Nurses’ attitudes are not independent of the general attitudes of society [7], and therefore, negative attitudes and myths about older adults are widespread, which must be compounded by the fact that health professionals are susceptible to age-related stereotypes due to the increased exposure of older people to disease [8]. Attitudes towards older people could affect the services provided to this age group and therefore change the value of the care provided by health professionals, and consequently their quality of life in older age [9,10,11,12,13]. Attitude can be defined as a person’s beliefs, feelings, and experiences about a human being, other animals, objects, or conditions, which are constituted by the person’s background, and which influence their behavior or reactions [14]. Many factors of care for older people could be related to the attitudes that healthcare worker groups have towards this population, including, for example, inadequate physical conditions and technical equipment in the workplace, lack of communication with care team professionals, and inadequate knowledge and/or skills for the care of older people [14]. In addition to these factors, culture, age, gender, level of education, experience, and previous relationships with older people could be influential factors regarding the attitudes of the healthcare provider towards the aging population.

At present, and following the COVID-19 pandemic, certain “ageist” stereotypes have been reinforced about care for older people. Thus, various international organizations have called on governments to protect the most vulnerable people, including older people. In the words of the World Health Organization (WHO) Regional Director for Europe, European countries, governments, and authorities must make interventions to ensure that all older people are treated with respect and dignity during these times without leaving anyone behind. Consequently, the use of scales to assess attitudes towards older people as well as knowledge of certain aspects of older people would help us to understand and model the appropriate strategy to be used for the training and support of professionals who provide care to these age groups. In this sense and given that most of the clients/patients that our future health professionals will have to care for will be older people, it is essential to know and carry out a study of the behavior and attitudes of nurses towards older people and the influential factors, as it is very important to improve practice standards and quality of care in the gerontological sector [15,16]. Likewise, educators must understand the factors that influence the attitudes of students toward the older people they care for. A clearer understanding of the attitudes that drive nurses towards the desire to work with older people would be a good starting point to encourage the development of positive and enriching attitudes [4,17,18].

Our current study is an investigation of our students’ attitudes toward caring for the elderly. Currently, we know how attitudes can influence the behavior of an individual, since people with a positive attitude will have more positive thoughts towards others; however, the current literature does not tell us what the causes of certain attitudes towards the elderly are nor how to solve them [19,20,21]. With our study, we want to know the attitudes of our students through a scale that collects most of the stereotypes towards these patients and from that basis undertake new studies that will redirect us to understanding how to introduce improvement actions in the quality care of the elderly patient. 

This study aimed to evaluate the psychometric properties of the Kogan Attitudes towards Older People Spanish version (KAOP-A) [22]) among Spanish undergraduate nursing students.

## 2. Materials and Methods

### 2.1. Participants and Procedure

The study population consisted of a convenience sample of 381 undergraduate university nursing students from Spain. Students were enrolled in the 2021–2022 academic year. The questionnaires were voluntarily completed using an electronic platform, and the anonymity of participants was ensured. The inclusion criterion was to be enrolled in the Ciudad Real School of Nursing, except for first-year students who had not yet experienced any type of contact with patients in hospitals or nursing homes. 

### 2.2. Instruments

A self-administered questionnaire was used to collect data. This questionnaire was designed by the researchers and based on the recent literature and had a total of 16 questions. Demographic characteristics were gathered, including both continuous variables (age) and categorical variables (gender, relationship status, level of education, religion, living with an older relative, etc.). In addition, information was collected on whether the students had contact in their families with elderly dependents or if they lived with them, as well as whether they had been in their hospital internships with elderly patient care services.

The Kogan’s Attitudes Towards Older People (KAOP) scale was developed by Nathan Kogan from the United States to assess nursing students’ attitudes toward aging [22,23]. This instrument has been widely used and translated into different languages, showing good validity and reliability to assess participants’ attitudes toward older people [10,23,24,25]. This instrument is composed of 34 items, half of which are positively worded, while the rest are negatively worded. All the questions in the questionnaire were included in the translation and validation of the questionnaire. The response options are given using a 6-point Likert scale, where 1 = strongly disagree and 6 = strongly agree. The scores range from 34 to 204, with higher scores meaning more positive attitudes toward older people. The Cronbach’s Alpha for the original scale was 0.81, the Spearman–Brown ranged from 0.66 to 0.83, and the inter-scale item correlation ranged from 0.46 to 0.52. The KAOP-Spanish version was translated from English to Spanish by two bilingual experts, following guidelines recommended for adapting cross-cultural instrument translations. In this study, Cronbach’s alpha coefficient for the total scale was found to be 0.75, indicating good internal consistency.

### 2.3. Ethical Considerations

This study was approved by the Ethics Committee of Castilla la Mancha University, according to the ethical guidelines established by the Helsinki Declaration in 2008 (C-294). Respondents participated anonymously and voluntarily in the research and written informed consent was obtained from all participants before enrollment.

### 2.4. Data Analysis 

SPSS 25.0 and JASP 0.16.2 were used for the statistical procedures. Construct validity, internal consistency, and reliability were assessed. For the internal construct validity, an exploratory factor analysis (EFA) with likelihood maximum and varimax rotation was performed. Kaiser–Meyer–Olkin (KMO) and Bartlett’s test of sphericity were calculated to identify the adequacy of the data and the sample size. The method to identify the number of factors was parallel analysis and only factorial loadings of over 0.30 were included. Cronbach’s alpha and the Omega coefficient were obtained for the reliability of the total scale and both factors (according to the original scale: prejudice and appreciation). 

## 3. Results

### 3.1. Demographic Data 

The demographic data of the participants are summarized in Table 1 (*n* = 381). There were 298 (78.2%) female and 83 (21.8%) male participants. The mean age of participants was 20.42 years (SD = 4.72), ranging from 18 to 56. Regarding the practical context of nurses, 52.5% worked with adults, 26.8% worked with older people, and 20.7% worked with kids. An estimated 21.3% lived in a rural environment, whereas 78.7% lived in cities. Up to 68.8% of participants did not have dependent older people in their family environment, 18.4% had at least one older person, 6.8% had two older people, and 6.1% had three or more. 

Descriptive results and item homogeneity. 

Table 2 includes the means and standard deviation of each item for the KAOP scale. In this regard, items of homogeneity were significantly low with a minimum value of 0.145 (item 1N) and a maximum value of 0.350 (item 11N). Several positive items (P) had higher mean scores than negative items (N).

### 3.2. Exploratory Factor Analysis and Reliability

Table 3 shows the factorial solution of the KAOP-S scale, factorial weights, proportion of variance explained, Cronbach’s alpha, and the Omega coefficient. The KMO was 0.78, indicating sampling adequacy (>0.50) and Bartlett’s test of sphericity was statistically significant at 3191.96, df = 561, *p* < 0.001. The factor analysis yielded a two-factor solution. The first factor was named “prejudice”, as in the original instrument. This factor was composed of negatively worded items. The second factor, called “Appreciation”, was composed of the positively worded items. The total explained variance was 25.69%, 14.48 for the first factor and 11.21 for the second one. 

Concerning reliability, Cronbach’s alpha for the total scale was 0.750 and the Omega coefficient was 0.949. For each factor, the findings were first factor (α = 0.776, Ω = 0.934); second factor (α = 0.773, Ω = 0.967).

## 4. Discussion

The main purpose of the study was to analyze the psychometric properties of the KAOP scale in a sample of undergraduate nursing students from Spain. 

The KAOP has proved to be an internationally effective instrument for assessing the attitudes of health professionals toward older people [10,24,26,27]. To the best of our knowledge, there is no validated version of this scale in Spanish. The Spanish version of the KAOP shows high reliability (internal consistency and stability) and good content and construct validity. The alpha coefficients were 0.75 for the total scale and 0.77 for both subscales. The Arabic version has the highest Cronbach’s alpha (0.89) [23,24] compared to the Indonesian version (0.70) [28], which has the lowest Cronbach’s alpha. The Greek (0.73) [26], the Swedish (0.79) [29,30], the Italian (0.76) [31], and the Spanish (0.75) versions have similar results, whereas the Chinese (0.82) [25], Turkish (0.85) [26], and Iranian (0.83) [32] versions obtained superior results.

The factor analysis of the KAOP revealed two well-defined factors, “prejudice” where the items describe a negative predisposition towards older people, and “appreciation”, representing positive items expressing positive feelings and opinions toward older people. Most versions include only two factors. Thus, the Chinese version reports two components with an explained variance of 54.7% [25], the Arabic version reports two factors with an explained variance of 60.12% [24], the Iranian version showed two factors with a variance of 58.76% [32], and the Turkish and Indonesian versions are based on two factors [26,28]. However, the Greek version is a six-component version with a variance of 41% [29], and in the case of the Swedish version, the ten-factor solution with a variance of 57.7% was not considered because of the difficulty of interpreting the data and the authors opted for a three-factor version with a variance of 30% [29]. In our case, the two-factor solution had an explained variance of 25%. Two negative items are included in the positive scale (3N and 7N) and one positive item (17P) is included in the negative scale. In the original scale, Kogan attempts to construct logically opposed meanings, however the experiences or feelings that these statements describe are not always psychologically opposed [31]. Thus, in our case, and probably due to cultural elements, items 3N and 7N are not items that express a negative reaction or attitude, but rather the opposite, because the fact that older people have a great influence on social life is not seen as something negative but rather as something positive. In the case of older people not changing, even though it is a stereotype, it is closely linked to traditional Spanish culture, which does not denote a negative attitude, but rather a stereotypical approach which should probably change so as not to clash with disease prevention guidelines in this age group where certain changes are possible. Along these lines, item 17P, which is also modified in the Italian version and is found in the factor “prejudice”, could be explained by the presence of negativity in the sentence.

Overall, the attitude of our students was pleasantly more positive toward older adults than negative. Our results are therefore similar to those found in the Turkish and Iranian validations and contrary to those published in the Jordanian, Swedish, and Chinese validations [26,27,28,29,30,31,32]. In relation to the items with the highest scores, it is striking that most of them revolve around the experience that older people bring with them because of the background that makes them wiser in certain aspects of their lives, the fact that they can have the same faults as people of different ages, that it is relaxing to be with them, and that it is good for older people to live integrated into the neighborhoods. Regarding the negative items, the highest scores fall on those aspects that have been stereotyped for many decades: the fact that it is very difficult to change their minds, that they complain a lot about the new generations, and that they constantly demand love. In our study, the majority of the sample claims not to live with dependent elders, which to some extent explains why the items relating to the independent activity of older adults have a higher score than the items claiming that older adults cannot be active in society. However, a previous study described closeness to older people as a factor of influence on positive attitudes toward older people [26,27,28,29,30,31,32].

Aging is accompanied by various changes in the form and function of internal and external organs, which can lead to poor adaptation to the environment among older people [32,33]. These changes are responsible for the fact that at some point the level of dependency in this age group is quite high, coupled with a higher life expectancy, thus, we are facing a greater demand for health care in this age group. In recent years, the figure of the “informal caregiver” (within the older person’s family), which is deeply rooted in Spanish culture [32,33], is beginning to disappear for several reasons. The family structure is starting to lack a residential nucleus, which is where all the members of the family used to live. In contrast, the different family members are beginning to live in distant towns, usually for work reasons, and new family models establish other roles for women, who have traditionally been the “informal caregivers” par excellence. In this context, the care of the older person in nursing homes must become a priority in the academic curriculum of our nurses as most of their patients/clients will belong to this age group [34,35]. Consideration of the needs and problems of older people is therefore an important social issue. Because of global ageing, the population’s health needs are increasing, warranting a higher proportion of health professionals. The quality of life of older people is an important topic because everyone has the right to experience healthy aging [36,37]. Consequently, to provide direct holistic care in the future, there will be a growing need for well-trained and committed people and nurses to work with older people in various settings, and especially more nurses with gerontological training to meet the needs of our older people [38,39]. In this sense, the first step to consider should be the positive attitude towards this age group, where stereotypes and workloads in certain nursing homes could contribute to an unhealthy work environment.

### Limitations

Firstly, we used a convenience sample of students in our faculty. We did not perform the test and re-test reliability analyses due to organizational reasons. Other measures, such as concurrent validity, could be performed in another sample to further support the validity of the KAOP-S because of the lack of another validated and reliable instrument for testing the attitude toward older people. Moreover, a confirmatory factor analysis should be applied with new samples to check the solution with two factors (prejudice and appreciation).

## 5. Conclusions

This study provides evidence that the Spanish version of the KAOP-S is a reliable and valid instrument for evaluating Spanish nursing students’ positive and negative attitudes toward older people. However, further testing in other populations such as professional nursing could be proven useful.

## Figures and Tables

**Table 1 healthcare-11-01321-t001:** Summary of participant gender, type of patients, and place of residence.

	n	KAOP Total	Prejudice	Appreciation
Male	83	112.59 (SD = 14.64)	51.14 (SD = 19.62)	61.44 (SD = 9.76)
Female	298	113.46 (SD = 13.60)	49.58 (SD = 10.07)	63.88 (SD = 9.65)
Work with Adults	200	111.70 (SD = 13.75)	49.29 (SD = 9.93)	62.40 (SD = 9.59)
Work with Elderly people	102	115.30 (SD = 13.40)	50.94 (SD = 10.08)	64.36 (SD = 9.63)
Work with Kids	79	114.65 (SD = 14.22)	50.21 (SD = 9.98)	64.44 (SD = 10.02)
Residence Rural	81	50.21 (SD = 9.98)	50.79 (SD = 10.72)	62.69 (SD = 11.31)
Residence Urban	300	50.21 (SD = 9.98)	46.69 (SD = 9.78)	65.53 (SD = 9.25)

**Table 2 healthcare-11-01321-t002:** Means, standard deviations, and item homogeneity for Kogan’s Attitudes towards Older People (KAOP).

**Items**	**M**	**SD**	**Skewness**	**Kurtosis**	**Item Homogeneity**
1N It would probably be better if most old people living in residential units with people their own age	3.69	1.314	−0.124	−0.820	0.145
1P It would be probably better if most old people lived in residential units that also housed young people	3.38	1.235	−0.075	−0.547	0.239
2N There is something different about most old people: it’s hard to figure out what makes them thick	3.13	1.254	0.217	−0.349	0.268
2P Most old people are really not different from anybody else: they are as easy to understand	3.44	1.256	−0.042	−0.454	0.276
3N Most old people get set in their ways and are unable to change	4.57	1.311	−0.814	0.287	0.252
3P More old people are capable of new adjustments when the situation demands it	3.06	1.238	0.432	−0.389	0.221
4N Most old people would prefer to quit work as soon as pensions or their children can support them	2.67	1.288	0.517	−0.282	0.209
4P Most old people would prefer to continue working just as long as they possibly can rather than be dependent on anybody	4.40	1.230	−0.458	−0.159	0.188
5N Most old people tend to let their homes become shabby and unattractive	2.52	1.285	0.591	−0.338	0.243
5P Most old people can generally be counted to maintain a clean attractive home	4.04	1.228	−0.250	−0.424	0.183
6N It is foolish to claim that wisdom comes with old age	2.72	1.398	0.552	−0.400	0.186
6P People grow wiser with the coming of old age	4.22	1.252	−0.383	−0.352	0.272
7N Old people have too much power in business and politics	3.03	1.098	0.176	−0.277	0.248
7P Old people should have more power in business and politics	3.23	1.122	0.171	−0.383	0.269
8N Most old people make one feel ill at ease	2.25	1.207	0.812	0.304	0.259
8P The elderly are relaxing to be with them	4.19	1.190	−0.370	0.032	0.261
9N Most old people bore others by their insistence on talking about the “good olds days”	2.37	1.249	0.769	0.047	0.281
9P One of the most interesting and entertaining qualities of most old people is their accounts of their past experiences	4.81	1.160	−0.693	0.085	0.246
10N Most old people spend too much time prying into the affairs of others and giving unsought advice	2.95	1.249	0.173	−0.495	0.253
10P The elderly mind their own business	3.10	1.191	0.318	−0.369	0.256
11N if old people expect to be liked, their first step is to try to get rid of their irritating faults	2.38	1.209	0.502	−0.378	0.350
11P When you think about it, old people have the same faults as anybody else	4.61	1.262	−0.704	0.123	0.207
12N In order to maintain a nice neighborhood, it would be best if too many old people did not live in it.	2.18	1.173	0.769	0.126	0.161
12P Neighborhoods are nice when integrated with the elderly	4.18	1.266	−0.389	−0.168	0.234
13N There are a few exceptions, but in general most old people are pretty much alike	2.96	1.236	0.245	−0.472	0.337
13P It is evident that most old people are very different form one another	3.95	1.309	−0.186	−0.655	0.151
14N Most old people should be more concerned with their personal appearance, they’re too untidy	2.55	1.148	0.417	−0.316	0.302
14P Most old people seem to be quite clean and neat in their personal appearance	3.72	1.168	−0.147	−0.338	0.310
15N Most old people are irritable, grouchy and unpleasant	2.35	1.164	0.698	0.186	0.217
15P Most old people are cheerful, agreeable and good humored	3.99	1.154	−0.362	0.048	0.347
16N Most older people are constantly complaining about the behavior of the younger generation	3.99	1.241	−0.268	−0.557	0.270
16P One seldom bears old people complaining about the behavior of the younger generation	2.65	1.204	0.748	0.215	0.243
17N Most older people make excessive demands for love and reassurance	3.63	1.327	0.010	−0.572	0.325
17P Most older people need no more love and reassurance than anyone else	2.39	1.387	0.834	0.020	0.195

Answers range = (1–6); N = negative items; P = positive items.

**Table 3 healthcare-11-01321-t003:** Factor loadings after varimax rotation for the KAOP scale.

	**Prejudice**	**Appreciation**
9N Most old people bore others by their insistence on talking about the “good old days”	0.660	
11N if old people expect to be liked, their first step is to try to get rid of their irritating faults	0.655	
14N Most old people should be more concerned with their personal appearance, they’re too untidy	0.654	
15N Most old people are irritable, grouchy and unpleasant	0.600	
8N Most old people make one feel ill at ease	0.586	
10N Most old people spend too much time prying into the affairs of others and giving unsought advice	0.583	
13N There are a few exceptions, but in general most old people are pretty much alike	0.542	
12N In order to maintain a nice neighborhood, it would be best if too many old people did not live in it	0.541	
5N Most old people tend to let their homes become shabby and unattractive	0.499	
2N There is something different about most old people: it’s hard to figure out what makes them thick	0.450	
4N Most old people would prefer to quit work as soon as pensions or their children can support them	0.440	
6N It is foolish to claim that wisdom comes with old age	0.392	
17P Most older people need no more love and reassurance than anyone else	0.352	
17N Most older people make excessive demands for love and reassurance	0.349	
16N Most older people are constantly complaining about the behavior of the younger generation	0.315	
1N It would probably be better if most old people lived in residential units with people their own age	0.188	0.118
16P One seldom bears old people complaining about the behavior of the younger generation	0.243	0.205
15P Most old people are cheerful, agreeable and good humored		0.667
8P Most old people would prefer to continue working just as long as they possibly can rather than be dependent on anybody.		0.609
12P Neighborhoods are nice when integrated with the elderly		0.594
9P One of the most interesting and entertaining qualities of most old people is their accounts of their past experiences		0.578
14P Most old people seem to be quite clean and neat in their personal appearance		0.564
11P When you think about it, old people have the same faults as anybody else		0.553
6P People grow wiser with the coming of old age		0.543
4P Most old people would prefer to continue working just as long as they possibly can rather than be dependent on anybody		0.537
5P Most old people can generally be counted to maintain a clean attractive home		0.491
2P Most old people are really not different form anybody else: they are as easy to understand		0.419
13P It is evident that most old people are very different form one another		0.414
7P Old people should have more power in business and politics		0.391
1P It would be probably better if most old peop0le lived in residential units that also housed young people		0.388
3N Most old people get set in their ways and are unable to change		0.360
10P Most old people tend to keep to themselves and give advice only when asked		0.322
3P More old people are capable of new adjustments when the situation demands it		0.310
7N Old people have too much power in business and politics		0.243
**Variance explained**	**14.48**	**11.21**
**Cronbach’s Alpha**	**0.776**	**0.773**
**Omega**	**0.934**	**0.967**

## Data Availability

Not available.

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
