# Peer review of "Adaptation and Validation of the Spanish Version of Kogan’s Attitude toward Older People Scale (KAOP)"

_healthcare, 2023, doi:10.3390/healthcare11091321_

Round 1

Reviewer 1 Report

Thank you for the opportunity to conduct this review.

This is an interesting article and I think it is very suitable for the profile of the journal in which it will be published.

The article is written and presented in an appropriate manner.

Congratulations for the work done.

 I have some recommendations for the author:

 The first section of the article (Introduction) is well designed and includes a comprehensive literature review.

WHO - please write without using an acronym - see line 69

 In the material and method:

-Please specify in the text how you determined the sample population included in the study.

 - Please specify what the inclusion and exclusion criteria were for the studiu.

-  I think it would be useful for the author to better describe the questionnaire in the Spanish version compared to the original version. For example total number of questions, how many questions were dedicated to the attitude scale, how many questions are dedicated to the knowledge scale? How many questions were dedicated to the demographic data collection part. Was there a scale of practices?

- As a reader, I would like to find out if the questions that make up the questionnaire were fully taken from the original questionnaire or only partially. What was the reason why some questions were not taken from the original questionnaire (differences in skills, lack of applicability, etc.)

 - I recommend the authors to specify what measures they used to limit the phenomenon of bias.

 Discussion

I suggest you start the " Discussion " section with the purpose of the study.

Author Response

File attached. 

Reviewer 2 Report

This is a clean and useful paper. My recommendations for improvement:

1) I think the paper would benefit from an expansion of the section ‘Discussion’. This type of paper/analysis is of interest to a variety of audiences, and I believe that it would be helpful to them if the authors added a paragraph to the Discussion, in which they describe and evaluate the results rather than just leave it to the numbers to do so. Do the authors find current students’ attitudes satisfactory? Is there room for improvement? Is there some attitude that stands out as particularly worrying or encouraging? Et cetera…

2) Line 57: “… experiences in relation to a human being, object, or condition…” The sequence (human, object) may unintentionally propagate the outdated, unscientific and ethically highly problematic view of sentient nonhuman animals as objects, ‘mere automata’. Given that the paper is centred on vulnerable populations (elderly humans), I believe a greater sensitivity to vulnerable populations more generally is warranted, and I’d like to see the authors add nonhuman animals between humans and objects: e.g. “in relation to a human being, other animal, object, or condition…”

Language: in a few places “from” is misspelled as “form”

Author Response

File attached

Reviewer 3 Report

 The research is very interesting where the authors conducted a study for the adaptation and validation of the Kogan Attitude Scale towards Parents (KAOP). This is motivated by the importance of understanding the behavior and attitudes of nurses towards the elderly to improve clinical practice and quality of care in the gerontology sector. A clearer understanding of the attitudes that drive nursing toward wanting to work with older people would be a good starting point for encouraging the development of positive and nurturing attitudes.

Abstract is good enough and structured, explaining the reasons why this research was carried out, but it is necessary to clarify what the aims and benefits of the research are, and it is better if the research method is explained explicitly and clearly.

Introduction, especially in the background section, the author needs to reinforce the arguments and gaps of the research, the formulation of the problem, the urgency of the research, and the aims and benefits of the research.

The research method used is actually good, namely through a cross-sectional study conducted among 381 nursing students at the Faculty of Nursing at the University of Castilla La-Mancha to evaluate the psychometric properties of the Spanish Version of the Kogan Attitude Scale towards Parents (KAOP-S). Construct validity, internal consistency and reliability were assessed. In total, 298 women and 83 men completed the questionnaire with their mean age being 20.42 years.

From the results of observations, it is very good where the authors show that the exploratory factor determines that the scale has a two-factor solution and an explained variance of 25% in the sample. KAOP-S was found to be a reliable and valid tool with good content and construct validity for assessing the attitudes of nursing students towards parents.

The table section needs to be explained properly and sharply so that it is even easier for the reader to understand.

References can be added with updated sources, both books, articles and journals. Thus strengthening the article more academically.

Author Response

File attached.
